# Consumer Participation in CSR: Spending Money versus Spending Time

**Yaping Fang** [1] , **Feng Liu** [2] , **Sunmin Kim** [3] **and Minchan Pyo** [3],*

1 Economics and Management School, Hefei University, Hefei 230601, China
2 Business School, Shandong University, Weihai 264209, China
3 Business School, University of Seoul, Seoul 02504, Republic of Korea
* Correspondence: minpyo@uos.ac.kr

**Abstract:** Consumer participation plays a more active role in corporate social responsibility (CSR) than ever before. However, a framework describing how participation approaches generate cognitive and behavior responses in consumers is still lacking. The purpose of this study is to investigate the different effects among consumers' participation approaches (i.e., spending money versus spending time) on inspiring consumers to engage in CSR. Additionally, we explore consumers' cognitive mechanisms by identifying the key mediating role of perceived value. A total of 429 participants were recruited using an inter-group between-subjects design, and hypotheses were tested by a structural equation model, including path analysis and bootstrapping procedure method. The results show that consumers tend to have a more positive perception of CSR and greater intention to participate when they spend time rather than money to engage in CSR activities. We also confirmed the importance of perceived value in CSR, as the link between consumer participation type and behavioral intention is fully mediated by perceived value. These findings shed a light on the importance of the participation approach in CSR, contributing to CSR and consumer participation research. Our study also provides meaningful implications for companies to encourage consumers to use their time to participate in CSR activities.

**Keywords:** CSR; consumer participation; time; money; perceived value; participation intention

## 1. Introduction

Corporate social responsibility (CSR) has become a critical business strategy for gaining a competitive advantage. Considering the important role of consumers' involvement or engagement in CSR activities, more and more businesses are tending to interact or co-create with consumers in the implementation process, which is referred to as consumer participation in CSR. For example, the Ant Forest program is a mini-application embedded in Alipay that focuses on tree-planting with consumer participation. According to publicly available information, 613 million consumers use Alipay every day to participate in the Ant Forest mini-program, resulting in the reforestation of 264,680 hectares and the plating of 326 million trees. Due to its contribution to protecting the environment, this program won the "Champion of the Earth for Inspiration and Action" award from the United Nations Environment Programme (UNEP). As a result, the Ant Forest program not only enhances corporate reputation, but it also greatly improves the customer retention and usage rates of Alipay, effectively countering the challenge posed by Tencent, one of the toughest competitors in the mobile payment market. The success of the Ant Forest program reaffirms that CSR with consumer participation can be an effective strategy for increasing consumer loyalty and shaping their awareness of CSR, which in turn gives companies a competitive advantage [1].

Corporate social responsibility (CSR) involving consumer participation has gained widespread attention and is actively implemented in practice. However, academic interest

in this issue remains limited [2]. While consumers can participate in CSR in various ways, determining which type of participation is most suitable for successful CSR is challenging. Previous research shows that consumers participating directly in CSR can result in positive marketing outcomes such as increased consumer trust and satisfaction [3], participation and purchase intentions [4,5], CSR associations, and credibility [6]. Additionally, CSR characteristic factors such as CSR participation effort [2,7], CSR domain (type) [8], message appeal and service type of CSR [9], CSR communication approach [10], and contribution type [11] are the antecedent variables that influence different consumer responses. Moreover, certain studies have examined the implementation region as another significant factor affecting the consumer's response towards CSR [12,13]. CSR participation type, as a CSR characteristic factor, may also affect consumer support and participation intention. However, current studies have not explored this issue in-depth. Previous studies mainly focus on the overall effectiveness of participatory CSR and the different dimensions of consumer cognition (relatively positive or negative), without empirically examining which types of participation are most suitable for inspiring consumers to engage in CSR. The underlying mechanism for how consumers decide whether or not to participate in a certain type of CSR also remains unclear. As a result, previous research does not provide clear guidelines for companies to explore effective participation approaches in CSR.

This study aims to provide a better understanding of the effectiveness of appropriate participation approaches in relation to consumers' intention to participate in CSR through empirical exploration. Individuals rely on two parallel and interacting information processing systems known as dual-process theory [14], which provides a theoretical framework for understanding the psychological mechanisms underlying consumers' decision-making. Previous research suggests that the activation of different modes in consumers during decision-making can be influenced by various factors including the concept of time and money; the concept of time tends to activate the emotional, holistic, and heuristic processing mode, while the concept of money activates the analytical and value maximization processing mode [14–18]. Research indicates that when compared to the concept of money, the activation of the concept of time would increase individuals' willingness to do good and donate more [15,19,20]. Furthermore, the inherent ambiguity of the theory of time, proposed by Okada and Hoch, highlights the intrinsic differences between money and time [21]. Consumers tend to perceive resources invested in terms of time as more flexible and ambiguous than those invested in terms of money, particularly with regards to opportunity cost assessment and perception of budget constraints [22–24]. This difference leads to varied perceptions and behavior towards CSR [21,24]. In this study, we divide CSR participation into two categories, spending money and spending time, to explore the impact of CSR participation on consumers' cognition and behavior in detail. Based on the aforementioned research findings of prior research, we suggest that consumers are more likely to have a positive perception of CSR and higher intention to participate when asked to invest time rather than money in CSR activities.

This study also examines the mediating variable that impacts consumers' decision about whether or not to participate in CSR. Schwartz identified several values that individuals universally pursued across the cultures [25,26]. Previous research demonstrates that perceived value directly influences consumer decision-making and behavior (i.e., [27]). Therefore, we identify personal values as a core element of the consumer participation mechanism in CSR. Specifically, we expect that the relationship between consumers' participation approach and their behavioral intention depends on how they perceive the value in CSR. In addition, we include consumers' perception of CSR as a dependent variable along with their participation intention. This is because CSR perception reflects the change in consumers' attitudes and opinions, while participation intention directly reflects the level of consumers' support for CSR. In summary, the purpose of this study is to explore the different effects of spending money and time on CSR activities and to identify the consumer cognitive pathway by exploring the key mediating role of perceived value between CSR type and participation intention.

To investigate the objectives of the study, an experiment using an inter-group design was conducted. Firstly, a pretest was conducted with 97 respondents to ensure that the study was appropriately designed. After the pretest, a paper-and-pencil questionnaire was used in the experimental study. The participants were recruited from libraries in China, and a total of 429 respondents participated in the survey. Descriptive statistics were obtained using SPSS 18.0, and a structural equation model was tested using AMOS 22.0 to assess the hypotheses.

This study proposes the following contributions. Firstly, few studies have recognized the significant role of consumers in CSR and identified the effective ways to implement CSR activities. Thus, we adopt the inherent ambiguity theory of time as a framework to categorize the CSR activities with consumer participation into two types: spending money and spending time. We apply the dual-process theory as a lens to explore the differential influence caused by CSR participation type, providing further theoretical evidence to understand the influence of consumers' participation in CSR. Secondly, we consider five perceived values (stimulation, security, hedonism, achievement, and universalism) simultaneously, following the suggestions of Green and Peloza [28]. They proposed that multiple values can be perceived simultaneously in CSR activities. This broader application of Schwartz's value classification aims to minimize the gap with previous research, which mainly focused on universalism or hedonism (i.e., [29–31], etc.). Finally, we suggest to practitioners that CSR activities with consumers' participation, especially those where consumers take time to participate, are an effective CSR strategy. Such activities can help establish a win–win cooperation partnership with consumers, cultivate company–consumer identification, and promote CSR co-creation behaviors [6].

In the following sections, we will review the related theoretical background, including CSR, consumer perceived value, and participation intention. Based on the literature, we will further develop our research hypotheses and conceptual models. Section 3 will present the study's research design, followed by an experimental study conducted in Shanghai, China, and the empirical results presented in Section 4. Finally, we discuss our findings and suggest implications, limitations, directions for future research, and conclusion.

## 2. Literature Review and Hypothesis Development

### 2.1. The Classification of Consumer Participation in CSR and Its Influence

Since the publication of Bowens' "*Social Responsibilities of the Businessman*" in 1953, Corporate Social Responsibility (CSR) has become a prominent topic in business, management, and marketing. CSR refers to the idea that companies should not only fulfill their corporate roles in society but also make efforts to address social issues that the government cannot solve [32]. Carroll and Buchholtz [33] believed that CSR is a less-imitable corporate strategy that can bring about favorable social outcomes, and it does not just imply the level of obligations for society or stakeholders. By implementing CSR, companies can strengthen their relationships with stakeholders [34], build corporate reputation [10], and improve financial and non-financial performance.

Consumer participation involves the discretionary expression of informational, emotional, physical, and behavioral contribution of consumer in the corporate service process [35]. In CSR, consumer participation is critical in supporting the company's CSR efforts associated with social issues (e.g., [1,2,7]). Consumer participation in CSR has been reported as an effective way to improve consumer trust and purchase intention and in turn gain a competitive advantage (e.g., [1,3,4]). Generally, consumers can participate in CSR activities by sharing CSR-related information with other consumers, answering their questions [3], or co-creating CSR with companies, such as planting trees together [1], which requires consumers to spend a certain amount of time. Additionally, consumers can express their support for a certain CSR by making monetary contributions (e.g., donating money directly).

This study distinguishes between two types of consumer participation in CSR: spending money and spending time. Spending money for CSR participation has been widely

studied and includes various forms, such as CRM (cause-related marketing) and charity donations (e.g., [33,36,37]). Spending time, on the other hand, involves completing certain missions that the company pledges to make donations or provide CSR support in return. For instance, in the Ant Forest mini-program, consumers must walk 100 days (10,000 steps per day) to plant a tree.

It is important to differentiate between the resources (i.e., time-to-money) invested by consumers in CSR because of their intrinsic differences in opportunity cost and budget constraints. This concept is known as the theory of inherent ambiguity of time [21,24]. While the opportunity cost of spending money is explicit and assessable because money can be easily converted and stored in the marketplace, time is non-transferable and cannot be saved for future use. Consequently, the opportunity cost of spending time is ambiguous. Additionally, while consumers have a fixed amount of disposable income during a specific period, budgetary constraints are often a reality. Nevertheless, time use is discretionary, and consumers can manage how they spend their time, even though everyone has the same 24 h in a day. These fundamental differences suggest that classifying participation types based on the resources invested by consumers in CSR, such as money and time, is reasonable. Furthermore, the different types of participation may have varying impacts on CSR-related perceptions and behaviors, which requires further investigation.

Research has shown that consumers perceive the time or money required to participate in CSR as both imposing costs and providing benefits [2,7]. The perceived costs are related to the monetary and/or non-monetary losses required for CSR participation. According to the theory of inherent ambiguity of time [21], these losses associated with spending time may be less significant than spending money. In fact, consumers may not realize the losses incurred by spending their time and may even enjoy it as leisure because both the opportunity cost and budgetary constraints of spending time are unclear. On the other hand, the perceived benefits of CSR participation include happiness and altruistic emotions related to positive value perception. Researchers have found that compared to spending money, consumers may derive more happiness (i.e., value perception), better CSR perception, and be more likely to increase social interaction and establish social connections from spending time [38,39]. Therefore, we believe that different types of participation in CSR may have varying effects and suggest that spending time has a more ambiguous perception of costs and a higher perception of benefits than spending money. The ambiguous perceptions of time costs may allow consumers to have better benefit perceptions, such as more value cognition and better CSR perception.

Moreover, this study suggests that spending time may elicit more positive behaviors from consumers than money. The dual-process theory suggests that individuals have two parallel and interacting systems (i.e., rational and experience system), and different information processing modes are activated when they make decisions in different situations [14]. Research has found that the experiential system is primed when consumers spend time, and the rational system is primed when spending money [14,40]. The rational system is based on the high level of consciousness, characterized by purposive, analytical, and describable traits. Individuals who rely on this system tend to make decisions based on cognitive processing. On the other hand, the experiential system is characterized by automation, integrity, association, and inexpressiveness. Individuals who rely on this system tend to make decisions based on intuition, emotion, and other irrational factors [41]. As mentioned earlier, time has the characteristics of ambiguity, difficulty in calculation and explanation, irreplaceability, and invisibility, which make consumers rely more on the experiential, emotional, and heuristic systems to process temporal information [14,15,18]. CSR as a good deed would trigger various positive emotions in consumers, such as happiness and a sense of responsibility [42]. Consequently, the willingness of consumers to participate in CSR would increase when the participation type involves spending time. However, money is more specific, easy to analyze, substitutable, and tangible, which makes consumers rely more on rational and analytic systems to process information related to money. It means that consumers would weigh the benefits against the costs of participating

in CSR through their cognitive process. When the outcome of the trade-off is negative, the willingness to participate may decrease or even disappear. Additionally, scholars have found that compared with money, activating consumers' concept of time can effectively improve individuals' participation intentions, such as willingness to donate to charity and increase the amount of donations [15,19,20]. Overall, spending time may elicit more positive value perception, and participation intentions from consumers than money. Hence, we propose the following hypothesis:

**H1.** *When the participation type is spending time, consumers' (a) perceived value, (b) perceived CSR, and (c) participation intention are relatively higher than when spending money.*

### 2.2. Consumer Perceived Value in CSR

In CSR, values are beliefs about what is important and desirable in an individual's life, which can be generated for all stakeholders including consumers [43,44]. Value is the basis of an individual's thoughts and behaviors [26], and each individual has their own unique set of values [25]. Consumer perceived value is generated from the interaction between consumers and products or companies [45], and it can be defined as "the overall assessment of the utility of a product based on the perceptions of what is received and what is given" [46]. When consumers are asked to participate in CSR, the assessment of the effectiveness of their participation will bring perceived value. Consumer perceived values in CSR are a multidimensional concept, such as utilitarian dimension (e.g., a high-quality product), emotional dimension (e.g., feeling good and happiness), or social dimension (e.g., obtaining social acceptance and self-esteem) [27]. Importantly, these values are not mutually exclusive; instead, they can be simultaneously perceived through a single CSR activity [28]. Therefore, comprehensively exploring value is crucial for understanding the role of each value in a particular CSR activity.

This study utilizes Schwartz's value theory [25,26] to examine the impact of consumer perceived value. Schwartz has identified values that are universally pursued across cultures, and these values can be classified into two dimensions: Self-Enhancement versus Self-Transcendence and Openness to Change versus Conservation. Self-Transcendence reflects concern for others' benefits or common welfare, which includes universalism and benevolence. Self-Enhancement, on the other hand, emphasizes an individual's personal success and power over others, which includes achievement and power. Openness to Change prioritizes an individual's independent thoughts and actions with an open mind, such as self-direction and stimulation, while conservation involves adhering to traditional practices and protecting stability, such as conformity, tradition, and security [26]. Hedonism is a special value which belongs to both Openness to Change and Self-Enhancement.

Previous research has confirmed the relevance of Schwartz's value theory in the CSR area, but these studies have only employed a few dimensions, such as universalism or hedonism (i.e., [29–31], etc.). In this study, all bipolar value dimensions are used to comprehensively investigate the impact of perceived value on CSR. Specifically, this study examines five values, including universalism, achievement, stimulation, security, and hedonism as the second-order variable to gain a broader understating of the relationship between consumers' participation type and their participation intention.

Given its significant influences on consumers' decisions and behaviors, the role of consumer perceived value in CSR has been widely studied [47]. Previous research has shown a positive correlation between consumer perceived value and perceived CSR. Specifically, consumers with high perceived value hold high expectations of CSR, including economic responsibility and ethical-philanthropic responsibility, and this perception can positively impact their perceived importance of CSR [48]. Moreover, consumer perceived values, such as hedonism, stimulus, achievement, security, and universalism, which are the focus of this paper, are critical for shaping CSR-relevant beliefs, preferences, and actions [30]. Additionally, a positive link between various benefits and consumer value perceptions (i.e., achievement, domination, change, growth, belonging, power, pleasure,

uniqueness) and their intention to participate has been confirmed [31,49]. Zasuwa [31] also claimed that perceived value is positively associated with behavior intention. Therefore, we propose that the perceived value has a positive impact on both perceived CSR and participation intention.

In addition to the direct effects mentioned above, perceived value may also mediate the relationship between consumer participation type and CSR perception and participation intention. However, previous research has not explored the mediating impact of perceived value in the relationship between CSR participation type and consumers' willingness to participate. This research gap needs to be addressed since perceived value is widely acknowledged as a key factor in CSR research [28,30,47–49]. Previous studies have suggested that CSR activities can generate various positive values (e.g., happiness, excitement) and lead to positive marketing outcomes, such as increased purchase intention, consumer support, and word of mouth [28,42,50]. Chen and Lin [51] also confirmed the mediating role of perceived value in increasing consumer satisfaction and participation intention in the context of social media marketing. Given the important role of consumers in both CSR with consumer participation and social media marketing, we believe that these arguments are equally applicable to our study and suggest that perceived value is a key mediator in participation type and consumers' positive responses. As previously noted, compared to spending money, spending time has a higher perception of benefits and a more ambiguous perception of costs. Therefore, we hypothesize that perceived value will have a higher mediating role between CSR and consumer responses (perceived CSR and participation intention) when consumers spend time (vs. money) to participate in CSR.

**H2.** *When the participation type is spending time, the mediating role of perceived value is higher in the process of influencing (a) perceived CSR and (b) participation intention than spending money.*

### 2.3. Perceived CSR and Participation Intention

Perceived CSR refers to the consumer's attitude or cognitive evaluation of a firm's CSR activity. Although CSR activities and perceived CSR are often used interchangeably, the former refers to the company's actual CSR initiatives, such as charity giving, while the latter is related to customers' awareness of the company's commitment to social responsibility and its associated activities [52]. Instead of "CSR activities", we will focus on "perceived CSR" in this study, because consumers' responses and behaviors are significantly influenced by how they perceive CSR [42]. For example, customers' CSR perceptions positively influence their purchase and loyalty intentions, which can give a company a competitive advantage [1,42]. Generally, CSR actions not only have a direct and primary impact on the focal company (e.g., consumer support and loyalty, resilience against negative information of company, and attribution of CSR motivation, etc.), but also have other outcomes for stakeholders (e.g., good partnerships with communities or non-profit organizations, reducing the stress caused by social issue). Consumers respond to certain CSR activities through various channels and approaches, and the aforementioned marketing returns depend on how consumers perceive CSR [42]. According to Ruiz de Maya [1], participatory CSR campaigns increase consumers' perceptions of CSR, leading to more positive outcomes.

Additionally, the relationship between consumer perceived CSR and participation intention has been studied extensively, with findings suggesting positive correlation between them. Hur et al. [4] found that consumer perceptions of CSR activities lead to greater CSR participation through the mediation of customer–company identification. Folse et al. [53] also verified the relationship between perceived CSR and participation intention through three studies, showing that consumers' CSR perception positively affects their participation intention in cause-related marketing. Similarly, Lee et al. [6], focused on the CSR communication leading to CSR participation and found similar results. Therefore, we expect that consumers' CSR perceptions can increase their intention to participate in CSR activities.

**H3.** *Perceived CSR positively affects consumers' participation intention.*

In summary, consumer participation type is related to consumer perceived value, perceived CSR, and participation intention in CSR. When the participation type is spending time, consumer perceived value (H1a), perceived CSR (H1b), and participation intention (H1c) are higher compared with when spending money. Perceived value mediates the relationship between participation type and perceived CSR (H2a) and participation intention (H2b). Furthermore, perceived CSR has a positive impact on participation intention (H3). Thus, we propose the following structural equation model (SEM) to demonstrate our hypothesis (Figure 1).

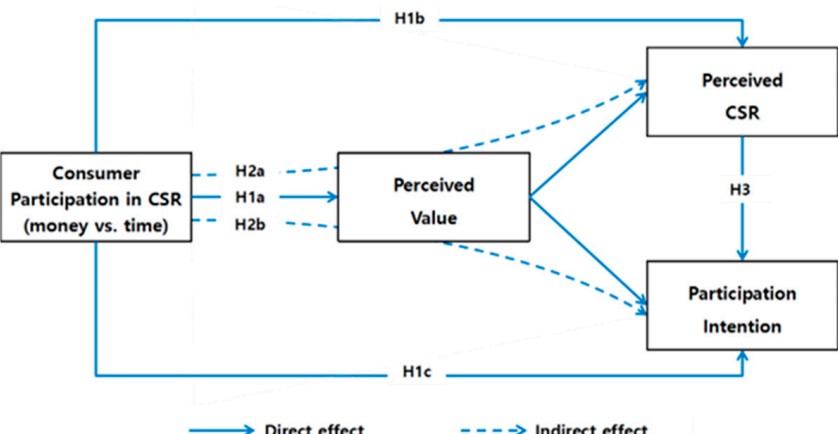

**Figure 1.** The proposed conceptual model. Notes: Participation type (spending money vs. spending time) in CSR is an observational independent variable, while perceived CSR and participation intention are two dependent variables. Perceived value is considered a first-order potential variable and is composed of five second-order potential variables: stimulation, security, hedonism, achievement, and universalism values. Attitude is included as a control variable in this model. All the measured items are provided in Supplementary material S2. The solid line represents the direct effect, and the dotted line represents the indirect effect.

### 3. Data and Methodology

#### 3.1. Experiment Design and Stimulus Development

This study employed an inter-group design to compare and analyze the impact of consumer participation type (spending money vs. spending time) on perceived value, perceived CSR, and participation intention in CSR activity. We developed an experiment stimulus, which included a brief corporate introduction and two promotional posters about an ongoing CSR activity that asked for consumer participation. The content of the two posters was identical except for the consumer participation type (spending money vs. spending time). In the experiment, respondents were randomly shown one of the two posters, and the influence of the participation type was explored by the score difference between the two groups.

In order to control the consumer's pre-existing preferences for real companies and to generalize the results, we created a fictious company for the study. To ensure the consistency between the experience group, we chose to focus on the same CSR activity—protection of the environment—as different CSR activities can elicit different cognitive response from consumers (e.g., [54]). To enhance the immersion in the experiment, we presented the CSR activity as an ongoing environmental protection effort actually taking place in China. The stimulus for the two groups differed, with one group being encouraging consumers to walk (spending time type), and the other groups being encouraged to make a purchase (spending money type). Except for the manipulation of experiment variables, the contents of both CSR activities were constructed equally. To enhance the realism of the poster, virtual logos, QR codes, and website addresses were also included. The details of the stimuli are provided in Supplementary material S1.

### 3.2. Pretest and Manipulation Check

To confirm that participants correctly identified the participation type shown in the poster, a pretest was conducted prior to the main experiment using the online survey platform Wenjuanxing (www.wjx.cn, accessed on 26 July 2022). The data were collected through snowball sampling and included 97 respondents (49 males and 48 females) aged 18 to 55 years (M = 28, SD = 6.65). Respondents were asked to select whether it takes ① money or ② time to participate in the CSR activity mentioned in the poster. The results confirmed that the manipulation was successful, with more respondents selecting "money" in the spending money stimulus and vice versa. Both groups were also familiar with each participation type asked about, and there was no statistically significant difference in familiarity between the two groups.

### 3.3. Sample and Data Collection Procedure

The experimental study used a paper-and-pencil questionnaire and was conducted under the approval of the Institutional Review Boards (IRBs) at the University of Seoul. To ensure semantic equivalence, we employed the back-translation technique for Chinese translation of the original questionnaires [55]. Additionally, face-to-face interviews with focus groups of Chinese graduate and PhD students were conducted to verify the comprehensibility of the questionnaire including scenarios and variable measures. During the survey process, we randomly distributed one of the two scenarios (see Supplementary material S1) to participants after confirming their willingness to participate in the experiment. Participants were asked to complete the questionnaire after reading the scenario.

Participants were recruited from libraries (i.e., Shanghai Municipal Library, Pudong Library, Xuhui Campus Library of Shanghai Jiao Tong University, Shanghai University of Science and Technology Central library) considering the convenience in data collection. A total of 429 respondents participated in the survey, with data from 411 respondents being used in the analysis after excluding 18 inadequate responses. The gender distribution was almost even (49.9% were women; n = 205). The average age of the respondents was 28 years, and more than 80% of the sample held four-year college degrees. Office workers accounted for 38.4% (n = 158) of the sample, followed by students (24.8%).

### 3.4. Measurement of Variables and Analysis Method

In this study, the consumer participation type in CSR was measured using a dichotomous variable, with a value of 0 indicating spending money and 1 indicating spending time. Except for the consumer participation type, all the other variables were measured using a 7-point Likert scale ranging from strongly disagree (1) to strongly agree (7).

The perceived value was measured through five dimensions: stimulation, security, hedonism, achievement, and universalism values, each consisting of three to four items based on [5,25,26,56]. For example, the concept of stimulation value is related with having interest, novelty, and challenging lifestyle, which was measured using three items (novelty, new, and exciting), while the hedonism which refers to personal happiness and sensory satisfaction was measured using three items (fun, pleasure, and enjoyment of life). Achievement value indicates an individual's perception of his or her own success in line with social standards, which was measured by four items (capable, influential, self-respect, and support). The security value is linked to the safety, harmony, and relationship stability, and three items (social order, health, and sense of belonging) were used to measure it. The universalism value is related to understanding, gratitude, and tolerance for nature and the environment, which was measured by three items (protecting the environment, a world of beauty, and unity with nature).

Perceived CSR refers to consumers' awareness of companies that fulfil commitments related to social obligations and activities [52]. The study included measurement items from [8,57–59], which were modified to fit the research context. Participation intention, which reflects the degree of consumers' willingness to engage or participate in CSR activities, was measured through five items adapted from the previous studies of [37,53].

Moreover, the method for delivering messages may affect the experimental results [9]. Hence, this study included attitude towards posters as a control variable in the survey to remove the bias from different expressions and sentence lengths in the two poster types. Four dimensions of attitude (dislike/like, negative/positive, bad/good, and unfavorable/favorable) [53] were measured.

For the analysis method, this study utilized structural equation modeling (SEM) to test our hypothesis as it provides a flexible framework for inferring cause-effect relationships among variables. Additionally, SEM enables the exploration of direct and indirect effects simultaneously. The path analysis and bootstrapping procedure method of SEM were focused on using AMOS 22.0. Furthermore, manipulation checks and descriptive statistics analysis were performed using SPSS 18.0.

## 4. Results

### 4.1. Manipulation Check and Descriptive Statistics

The results of the manipulation check showed that the two different types of CSR participation (spending money vs. spending time) led to significantly different perceptions among consumers. Participants who were assigned to the spending money condition perceived that more money was needed to participate in CSR activities; whereas those assigned to the spending time condition recognized the need for more time to participate. These differences in perception between the two groups, as measured by mean scores, were statistically significant, which was shown in Table 1. This demonstrates that the manipulation was successful. Additionally, the mean values of all variables were over four, indicating a positive attitude toward CSR with consumer participation. The collected data were found to be normally distributed based on skewness and kurtosis test results presented in Table 2.

**Table 1.** Manipulation Check.

| Question | Participation Types | Mean | SD | *t*-Value |
|---|---|---|---|---|
| Money is needed | Spending money poster | 4.75 | 1.79 | 7.54 *** |
| | Spending time poster | 3.48 | 1.64 | |
| Time is needed | Spending money poster | 4.76 | 1.75 | −4.66 *** |
| | Spending time poster | 5.51 | 1.51 | |

*** $p < 0.001$.

**Table 2.** Descriptive Statistics.

| Construct | Mean | SD | Skewness | Kurtosis |
|---|---|---|---|---|
| Stim | 4.94 | 1.55 | −0.798 | 0.130 |
| Secu | 4.97 | 1.40 | −0.564 | 0.005 |
| Hedo | 5.01 | 1.38 | −0.635 | 0.284 |
| Achi | 5.10 | 1.39 | −0.930 | 0.964 |
| Univ | 5.66 | 1.35 | −1.240 | 1.712 |
| PI | 4.68 | 1.36 | −0.527 | −0.218 |
| CSR | 4.99 | 1.15 | −0.623 | 1.320 |
| AT | 5.30 | 1.36 | −0.772 | 0.444 |

Notes: *Stim* stimulation, *Secu* security, *Hedo* hedonism, *Achi* achievement, *Univ* universalism, *CSR* perceived CSR, *PI* participation intention, *AT* attitude.

### 4.2. Common Method Bias and Validation of Measures

Harman's single-factor test was used to check for common method bias (CMB), and the proportion of the first principal component in the unrotated factor matrix was 45.94%, indicating that the CMB risk was low in the dataset. In addition, confirmatory factor analysis (CFA) was used to identify inappropriate items and enhance the operationalization of instruments, including the assessment of convergent and discriminant validity. The model fit appears to be acceptable with $x^2 = 372.203$, df = 138, $x^2/df = 2.697$, GFI = 0.916,

RMSEA = 0.064, NFI = 0.936, CFI = 0.959, TLI = 0.949. Convergent validity refers to the level of consistency between the potential and observation variables when measuring potential using them. As shown in Table 3, all factor loadings were higher than 0.7, average variance extracted (AVE) was higher than 0.67, and composite reliability was higher than 0.81, indicating convergent validity. In addition, the value of Cronbach's $\alpha$ was higher than 0.8, revealing satisfactory reliability. The analysis results showed that the square roots of the AVE were higher than the correlations between the latent variables, which confirmed discriminant validity (Table 4). Overall, these results suggest that our model is suitable for assessing this study's four hypotheses.

**Table 3.** Reliabilities and Convergent Validities.

| Construct | | Factor Loading | Cronbach's $\alpha$ | C.R. | AVE |
|---|---|---|---|---|---|
| Stim | Stim1 | 0.941 | 0.891 | 0.894 | 0.808 |
| | Stim2 | 0.855 | | | |
| Secu | Secu2 | 0.889 | 0.865 | 0.865 | 0.762 |
| | Secu3 | 0.857 | | | |
| Hedo | Hedo2 | 0.879 | 0.923 | 0.926 | 0.863 |
| | Hedo3 | 0.976 | | | |
| Achi | Achi1 | 0.858 | 0.814 | 0.816 | 0.689 |
| | Achi3 | 0.801 | | | |
| Univ | Univ2 | 0.889 | 0.912 | 0.894 | 0.808 |
| | Univ3 | 0.909 | | | |
| PV | Stim | 0.796 | 0.924 | 0.911 | 0.672 |
| | Secu | 0.762 | | | |
| | Hedo | 0.833 | | | |
| | Achi | 0.911 | | | |
| | Univ | 0.788 | | | |
| CSR | CSR1 | 0.887 | 0.899 | 0.902 | 0.755 |
| | CSR2 | 0.934 | | | |
| | CSR3 | 0.778 | | | |
| PI | PI1 | 0.779 | 0.831 | 0.837 | 0.721 |
| | PI2 | 0.914 | | | |
| AT | AT1 | 0.823 | 0.883 | 0.884 | 0.718 |
| | AT2 | 0.847 | | | |
| | AT3 | 0.871 | | | |

Notes: *Stimu* stimulation, *Secu* security, *Hedo* hedonism, *Achi* achievement, *Univ* universalism, *PV* perceived value, *CSR* perceived CSR, *PI* participation intention, *AT* attitude.

**Table 4.** Discriminant Validities and Correlations.

| Construct | PA | PV | CSR | PI | AT |
|---|---|---|---|---|---|
| PA | - | | | | |
| PV | 0.135 ** | **0.820** | | | |
| CSR | 0.051 | 0.713 *** | **0.869** | | |
| PI | 0.012 | 0.695 *** | 0.472 *** | **0.849** | |
| AT | −0.033 | 0.650 *** | 0.500 *** | 0.562 *** | **0.847** |

Note: *PA* participation approaches, *PV* perceived value, *CSR* perceived CSR, *PI* participation intention, *AT* attitude. The bold numbers are the square root of AVE, ** $p < 0.01$ *** $p < 0.001$.

### 4.3. Structural Equation Model and Hypothesis Tests

Before testing the hypotheses, the SEM's overall fit was assessed, and the goodness-of-fit indices in Table 5 revealed a satisfactory model fit: $x^2$ = 372.203, df = 138, $x^2$/df = 2.697, GFI = 0.916, RMSEA = 0.064, NFI = 0.936, CFI = 0.959, TLI = 0.949. The direct effects of consumer participation type on perceived value, perceived CSR, and participation intention were observed through a path analysis. As illustrated in Table 5, consumers' perceived value is higher in the case of spending time participating in CSR than in the case of spending money ($\beta$ = 0.157, $p < 0.001$), verifying H1a. However, consumer participation type does not directly influence perceived CSR ($\beta$ = −0.040, $p > 0.05$) or participation

intention (β = −0.064, *p* > 0.05), which rejects H1b and H1c. Additionally, perceived value directly affected perceived CSR (β = 0.682, *p* < 0.001) and participation intention (β = 0.634, *p* < 0.001), providing evidence to testing H2. Contrary to expectations, the effect of perceived CSR on participation intention was not statistically significant (β = −0.067, *p* > 0.05), thus rejecting H3.

**Table 5.** Standardized Parameter Estimates in the Structural Equation Model.

| | Model Paths | Estimates | S.E. | *t*-Values | *p*-Values | Results |
|---|---|---|---|---|---|---|
| H1a | PA -> PV | 0.157 | 0.116 | 3.656 | <0.001 | Accepted |
| H1b | PA -> CSR | −0.040 | 0.094 | −0.984 | 0.325 | Rejected |
| H1c | PA -> PI | −0.064 | 0.115 | −1.501 | 0.133 | Rejected |
| H3 | CSR -> PI | −0.067 | 0.078 | −0.991 | 0.321 | Rejected |
| | PV -> CSR | 0.682 | 0.060 | 9.719 | <0.001 | Providing |
| | PV -> PI | 0.634 | 0.093 | 6.735 | <0.001 | evidence |
| | AT -> PV | 0.655 | 0.057 | 11.423 | <0.001 | for H2 |
| | AT -> CSR | 0.056 | 0.051 | 0.917 | 0.359 | |
| | AT -> PI | 0.182 | 0.063 | 2.820 | 0.005 | |

| | | Goodness-of-fit indexes | | | | |
|---|---|---|---|---|---|---|
| $x^2$ (138) | $x^2$/df | GFI | RMSEA | NFI | CFI | TLI |
| 372.203 | 2.697 | 0.916 | 0.064 | 0.936 | 0.959 | 0.949 |

Notes: *PA* participation approaches, *PV* perceived value, *CSR* perceived CSR, *PI* participation intention, *AT* attitude.

The bootstrapping procedure method was used to verify the mediating effect of perceived value by confirming whether the bootstrapped confidence intervals (CIs) were different from zero. We performed bootstrapping by setting 5000 bootstrap samples at a level of 0.05. The analyses revealed evidence of mediation (Table 6). The direct effects, H1b and H1c, were rejected; therefore, the bootstrapping results explained that perceived value played a complete mediating role in the process of influencing consumers' perception of CSR (β = 0.107, 95% CI [0.048, 0.178]) and their participation intention (β = 0.095, 95% CI [0.041, 0.164]), showing that H2a and H2b are supported.

**Table 6.** Mediation Effects of Perceived Value.

| | Path | β | S.E. | *p* | 95% Bias-Corrected Bootstrap CI | | Results |
|---|---|---|---|---|---|---|---|
| | | | | | Lower | Upper | |
| H2a | PT -> PV -> CSR | 0.107 | 0.032 | <0.001 | 0.048 | 0.178 | Accepted |
| H2b | PT -> PV -> PI | 0.095 | 0.031 | <0.001 | 0.041 | 0.164 | Accepted |

Notes: *PT* participation types, *PV* perceived value, *CSR* perceived CSR, *PI* participation intention.

## 5. Discussion

### 5.1. Main Findings

This study investigates the impact of two types of consumer participation (i.e., spending money and spending time) on consumers' cognitive and behavioral responses, and the results verified that spending time is the more effective strategy to encourage consumers to participate in CSR directly. Firstly, we found that when consumers participate by spending time, their perceived value is higher compared to spending money. Specifically, investing time in CSR evokes positive feelings (e.g., sense of belong, which is security value) and emotions (e.g., happiness and existing, which are hedonism value and stimulus value), and enhance the sense of personal achievement (i.e., achievement value) and environment protection awareness (i.e., universalism value) more than investing money. Secondly, the study found that the relationship between consumer participation type and participation intention is fully mediated by perceived value. Through the mediating role of perceived value, when consumers participate by spending time, they tend to have a more positive

perception of CSR and are more likely to participate in CSR. These findings reinforce the importance of value in CSR. Interestingly, we also found that a positive CSR perception may not always lead to positive behavioral intentions, contrary to previous studies that found a positive relationship between consumers' perceived CSR and participation intention [53]. This phenomenon is known as the CSR dilemma or paradox in the CSR communication literature [10], where the more companies fulfill their CSR, the more consumers doubt their intentions and hesitate to participate in CSR activities.

*5.2. Theoretical Contributions and Implications*

This study offers valuable theoretical contribution and implication for academics. Firstly, while many businesses engage consumers in their social activities, academic research in this area is still limited. Our study fills this gap by examining the different effects of consumer participation types (spending money vs. spending time) on cognitive and behavioral responses to CSR. Our findings reveal that when consumers participate through spending time, their perceived value and participation intention are higher compared to when they participate through spending money. This supports Howie et al.'s [2] suggestion that different types of participation lead to varying cognitive and behavior responses. However, our study expands on this by directly examining which types of participation are more effective in inspiring consumers to engage in CSR and revealing the mediating role of perceived value in this relationship.

Secondly, we contributed to CSR literature by extending the application of two important theories: the inherent ambiguity theory of time and dual-process theory. We categorize consumer participation in CSR based on the inherent ambiguity theory of time, providing meaningful insights for expanding previous studies on CSR classification. Furthermore, previous research [11,54] has interpreted similar findings to ours as consumer psychological biases, where non-monetary contributions are perceived as purer, involve greater effort, and elicit more emotionality than monetary contributions. Our study provides a theoretical framework to support these findings by suggesting that these psychological biases are caused by the inherent ambiguity of time. Additionally, we apply dual-process theory to explore the differential influence of CSR participation type. To our knowledge, this is the first study to apply dual-process theory in CSR literature, promoting the development of CSR research at the individual level. In summary, this study provides theoretical evidence to understand the CSR participation types of consumers and explores the application of the inherent ambiguity theory of time and dual-process theory in CSR literature. The implications of our study can assist businesses in developing more effective CSR strategies and can guide policymakers in promoting socially responsible behavior among consumers.

Thirdly, we have confirmed the mediating effect of perceived value, building upon previous studies that have highlighted its existence in CSR [27–29]. Our research goes further, establishing perceived value as a key driver that influence both consumers' perceptions and behavioral intentions in CSR. Factors such as happiness, uniqueness, sense of belonging, achievement, and environmental conscientiousness not only enhance consumers' positive impressions of the focal company but also directly impact their participation intention in the activity itself. To determine how participation approach can be linked to participation intention, we applied Schwartz's value classification [25] extensively. Unlike prior research, which focused on specific values such as universalism [30,31] or hedonism [29], our empirical results indicate that consumers can perceive multiple values simultaneously in CSR activities, confirming Green and Peloza's previous argument [28].

Finally, this study provides some insight for Ahn and Lee [7], who claimed that consumers' participation in CSR serves two roles: providing the "warm glow" and imposing perceived costs. First, the theories applied in this study, such as Schwartz's value theory and the theory of inherent ambiguity of time, provide the theoretical framework for their findings. Specifically, the role of providing "warm glow" could be measured based on Schwartz's value theory [25,26], and the role of imposing perceived costs may come from the theory of inherent ambiguity of time [21]. "Warm glow" refers to the moral satisfac-

tion or perceived benefits obtained from pro-social behaviors, and consumers perceive CSR participation as providing various benefits, such as happiness and social connections, which are related to perceived value [2,7]. Schwartz identified 10 universally pursued values across the cultures, providing an empirical measurement scale for "warm glow". The process of evaluating these monetary and/or non-monetary losses required in CSR participation may need the help of the theory of inherent ambiguity of time [3]. The uncertainty of time makes consumers more flexible and ambiguous in assessing the cost of spending time. Once consumers take time to participate in CSR activities, they may not realize the losses caused by spending time. Consumers will unconsciously lower their expectations about CSR, and therefore, they may feel higher satisfaction with the outcomes of participation. In addition, the results of this study provide empirical support for their statement. We found that all the scores of values are over 4.9 (Table 2), which means that consumers have positive value perceptions of CSR no matter what the participation type is. This is the role of providing the "warm glow" in CSR caused by consumer participation. We also found that when the participation type is spending time, the effectiveness of CSR is better than spending money. This reflects the difference in the imposed cost perception by the different information processing modes primed by time and money.

*5.3. Managerial Implications*

This study offers several managerial implications for practitioners. First, companies and managers should pay more attention to consumer participation approaches when designing the CSR activities involving consumer participation. Consumers may respond differently to different participation approaches. Our results indicate that spending time is a better option than spending money to encourage consumer participation in CSR activities. This provides companies with an opportunity to identify appropriate CSR activities to implement. Compared to monetary contribution, consumers' time investment in CSR activities could elicit positive feelings and emotions, enhance personal self-achievement and self-expression, and consequently result in more positive CSR perception and behavioral intention. Practitioners need to focus on the consumer participation approach in CSR activities and avoid relying solely on asking consumers to donate money or purchase corporate products. Instead, companies should encourage consumers to take some time to participate in CSR activities. By doing so, the motivation for CSR would be purely recognized, the CSR would be considered more valuable, and participation intention would be significantly increased.

Furthermore, to enhance consumers participation in CSR activities, companies should aim to optimize the perceived value of such activities; this involves designing strategies that cater to customers' intrinsic needs (such as feeling good and happy) and extrinsic aspirations (such as staying fit, saving money, and protecting the environment). Consumers typically adopt a "utilitarian" perspective when it comes to participating in CSR activities, prioritizing the expected value and rewards they stand to receive. To encourage greater consumer engagement, companies should focus on maximizing the perceived value and benefits of CSR activities while minimizing perceptions of cost. One effective strategy to achieve this is for companies to consider what consumers stand to gain from participating in CSR activities. In comparison with other marketing variables associated with CSR activities, perceived value is generally easier to measure.

A cautious approach should be taken towards companies that use CSR as a corporate strategy. Our findings suggest that massive donations or other contributions by the company may not always positively encourage consumers to participate in CSR. Instead, advocating or publicizing a company's charitable behavior may activate consumers' persuasion knowledge and be recognized as a marketing tactic for profit. This is known as the CSR paradox in the field of communication which refers to the phenomenon that the more CSR is performed, the more skeptical consumers become about its motivation and effects, discouraging them from participating in CSR activities. By combining our findings with prior research, companies can use diverse strategies to enhance their CSR campaigns.

For instance, they can utilize the Social Networking Services to optimize their interactive capabilities and entice consumers to engage in CSR activities by leveraging "likes" or recommendations from other consumers. Another approach is to "gamify" CSR activities to stimulate consumer participation.

*5.4. Limitations and Future Research*

Although this study provides important insights into CSR, there are several limitations that should be discussed in future research. Firstly, the experiment used a fictitious beverage corporation. In future research, it is necessary to replicate this study using an authentic company to enhance the external validity. Additionally, future research should explore this empirical method in other regions and counties to identify differences across cultural backgrounds. Furthermore, this study only examined the environmental context. Therefore, it would be valuable to conduct various studies in different CSR activities, such as supporting low-income individuals, preserving cultural relics, and protecting animals, to generalize our findings. Besides, it is important to consider how the frequency of requests for consumer time investment affects consumers' cognition or willingness to participate in CSR activities.

## 6. Conclusions

Consumer participation is crucial in CSR, and this study aimed to investigate the impact of spending money versus time and the underlying mechanism of consumers' participation in CSR. Our findings suggest that spending time is a better solution than spending money due to the mediating role of consumer perceived value. Consumers' time investment in CSR could result in a positive value perception such as happiness, uniqueness, sense of belonging, achievement, and environmental conscientiousness. This perception could improve CSR perception and participation intention. Therefore, companies should consider consumers' participation approaches and perceived value as significant determinants of their participation behaviors in CSR.

**Supplementary Materials:** The following supporting information can be downloaded at: https://www.mdpi.com/article/10.3390/su15075786/s1.

**Author Contributions:** Conceptualization, investigation, methodology, writing—original draft preparation, project administration, Y.F.; writing—review and editing, Y.F., M.P., F.L., S.K.; supervision, M.P. All authors have read and agreed to the published version of the manuscript.

**Funding:** This work was supported by 2022 University Science Research Project of Anhui Provincial Department of Education (Grant Nos. 2022AH051769) and the Talent Research Fund Project of Hefei University (Grant Nos. 20RC65).

**Institutional Review Board Statement:** This study was approved by University of Seoul Institutional Review Board (IRB).

**Informed Consent Statement:** Informed consent was obtained from all subjects involved in the study.

**Data Availability Statement:** The datasets used and analyzed during the current study are available from the corresponding author on reasonable request.

**Conflicts of Interest:** The author declares no conflict of interest. The funders had no role in the design of the study, in the collection, analyses, or interpretation of data, in the writing of the manuscript, or in the decision to publish the results.

## Abbreviations

| | |
|---|---|
| Achi | achievement |
| AT | attitude |
| CRM | Caused-related marketing |
| CSR | perceived CSR |
| Hedo | hedonism |
| PA | participation approaches |

PI        participation intention
PV        perceived value
Secu        security
Stimu        stimulation
Univ        universalism

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
