# Peer review of "Consumer Participation in CSR: Spending Money versus Spending Time"

_sustainability, doi:10.3390/su15075786_

Round 1

Reviewer 1 Report

Dear Author,

1) Please revisit your abstract to provide a complete summary of your study, including the Objective of the study; Methodology; Findings; Conclusion; and Implication. Most of the sentences need rephrasing and English editing.

2) It is not clear what is the research gap in this area of study. The author can explain what is the research gap for this study. 

3) Research design and approach require further elaboration.

4) The list of abbreviations used in the article should be in the appendix.

5) References should follow the guidelines of the journal.

6) This manuscript needs to be proofread.

Reviewer 2 Report

The originality of the paper lies in its research context. However, the authors must clearly state the gap in the literature by supporting their arguments with previous studies.  

Authors missed some recent articles investigating CSR, I recommend reading them and using them as references:

Alawamleh, M. and Giacaman, S. (2021), "Corporate social responsibility impacts on Palestinian and Jordanian consumer purchasing", International Journal of Organizational Analysis, Vol. 29 No. 4, pp. 891-919.

Alizadeh, A., 2022. The Drivers and Barriers of Corporate Social Responsibility: A Comparison of the MENA Region and Western Countries. Sustainability, 14(2), p.909.

After great work authors decided to ignore this part (there is no discussion nor implications) these must be added. 
Discuss your results and compare them to LR.
Show how organizations can benefit from your research.
This is very important.

Reviewer 3 Report

English language needs thorough editing and correction; there are many problems with language throughout, sometimes making the points hard to follow. The authors would be well advised to get a good editor to correct the English and improve understandability of the study.

Although there may be a kernel of interesting material in the empirical part of this paper, the conceptualization of the study is very problematic, the model is insufficiently developed and not explained at all, and no discussion of the data gathered/analysis process is provided. Very little explanation of variables or relevant theories is provided, and there is little justification of the need for the study (or even explanation of why studying consumer participation is important. Definitions of concepts (e.g., consumer participation in CSR) need to be provided, and explanations of theories when they are introduced—along with some sort of rationale for using that particular theory (other than it might have been done before)—provided. Thus, conceptualization, argumentation, and literature support all need considerable work.

Argumentation/development of hypotheses: There is little real argument or explanation for the first hypothesis offered (given lack of definitions and logical development of ideas). Arguments for the second hypothesis are somewhat better handled, and some definitions are provided, though you still need to make the link to CSR and why particular values might be associated with different CSR participation behaviors. But...again, what is ‘high perceived value’ and perceived CSR—is that the same thing? In short, because there is insufficient definition of core concepts/constructs and little real argumentation, the rationale for the hypothesis is also inadequate and needs further development.

The model in Figure 1 is simply inserted into the paper without explanation of the relationships among the different variables—or why and how they matter.

Methods: Because the variables are not well explained and the relationships among them are unclear, it is not entirely obvious why these variables were selected. For example, where in the model do Schwartz’s values lie? What actual data were gathered? Even in the methods section, it is not clear why ‘time’ and ‘money’ (are measured as they are...seems too dichotomous) or what they are being related to—or how (and why) think like to Schwartz’s values. Better arguments up front would help here and in the results discussion.  

Results: Since I am not a quantitative scholar, I cannot comment much on the tests that were undertaken. But why was a factor analysis performed? Since the data gathered are not clearly articulated, it is hard to understand what the findings actually mean—or whether they have much meaning at all.

Specific comments:

p. 2: what is a ‘warm glow’ with respect to consumer participation?

p. 2: please explain why it matters what type of participation inspires consumer to engage in CSR. And what does ‘engaging in CSR’ even mean?

p. 2: You need to introduce your study more directly and explicitly without reference to the particular theories, but providing an explanation of what you are doing and why—and, essentially, making an argument for why your study is needed. E.g., when introducing ‘inherent ambiguity theory of time’, ‘dual process theory, please explain what these theories are briefly, then explore them more fully in the literature review. But you need here to provide some rationale for using these theories in the first place. Similarly, when introducing your ‘mediating variables’ from Schwartz’s theory. In other words, your paper needs to be about what you are studying not random theories thrown together. The use of theories/frames should support the study you have undertaken and provide insight into why the particular variables studied have been selected—so that they can answer the main questions you are asking.

It is not clear what the sentence that introduces these ideas is trying to say. You need a far better explanation of the point of your study here that actually explains what you are attempting to understand. Be basic: assume that your reader knows nothing about what consumer participation is, what different types exist, and why it would matter how they view or ‘participate in’ (whatever that means...) CSR.

P.3: maybe consider moving up (and elaborating) the ‘purpose of this study’ sentence to the front of the paper—partly to address the above concerns (and, again, using the theories as support, not putting them first).

p.3: Again, what is the ‘inherent ambiguity theory of time” and why is it relevant to consumer participation in CSR. What do you mean by ‘consumer participation in CSR’?

p. 3: there is a vast literature on CSR, and numerous definitions—you need to grapple with at least some of that literature and come to a more definitive definition of how you are using the term (which actually Crane and Matten, cited, do provide).

Similarly with consumer participation, when you get around to defining it. The definition is so vague as to be meaningless. In practice, what does such involvement look like? Give an example to ground this idea.  

Since most companies don’t seem to think or do much about ‘consumer participation’ when implementing their CSR programs, the idea that such participation is a ‘critical support’ (with no citations to support it), seems overstated. How, for example, does providing information about brand preferences or interacting with other consumers constitute participation in a company’s CSR?

p.4: What is the ‘inherent ambiguity of time’ (again)? Whose theory is it? Why have you selected this theory? What is the link to CSR or participation? (I could go on, but you get the point here: make an actual argument that supports the need for your study and support it with relevant literature).

p. 5-7: similar comment as above with ‘dual process theory. Explain it and provide a rationale why it is important to use it for this study. Also what is ‘consumer perceived value’? How does that relate to Schwartz’s five values—and link to CSR behaviors? What is the argument for a mediator role vs a direct role for perceived value for H2?

p. 7: Ah, p. 7 is where you define perceived CSR...but how does that relate to perceived value?

p. 7 ff: Am not going to comment in detail given the nature of  my overview comments above.

Reviewer 4 Report

This is an interesting paper, making a sound contribution to existing research, with its focus on consumer participation in CSR activities.

The paper is making a broad range of theoretical claims based on one set of experiments. In particular, the proposed contribution in terms of the work of Ahn and Lee needs further justification, to clarify that it is supported by the findings of the research.

Some consideration needs to be given to how time on a one off basis versus multiple requests for time may influence behaviour. That is, while people may be willing to invest time on a one off basis, their willingness may decline with repeated requests for investment of time. I realise this is not the focus of the paper, but it could be mentioned as an area for future research.

In 5.4, the authors suggest the results should be generalised with caution. Based on the approach, the point should be made rather that the results can not be generalised, but provide insights that could be examined in the context of other industries and in other national contexts.

The written expression and grammar requires extensive attention to ensure the authors are communicating their argument as effectively as possible.

Round 2

Reviewer 2 Report

this is much better than the previous version. I recommend accepting this manscript.

Reviewer 3 Report

The paper is much improved, and the authors have responded well to my previous comments, though it still needs a careful proofreading by a Native English speaker, particularly around the use of articles and grammatical issues like subject-verb agreement, use of singular words where plurals are needed, some word choices that need fixing (participation vs participant, value vs values), and the like. That said, the study is now quite comprehensible. Explanations are considerably clearer and good definitions, supported by literature, are provided for key constructs. The methods are now well explained, and the model is also explained. The Main Findings section particularly needs grammatical/English editing attention.

Again, I cannot attest to the validity of the quantitative work here, as I am not a quantitative scholar. Explanations are sound, and the rewritten conclusion is much more impactful than the original.